# Socioeconomic Patterning of Stunting and Overweight Among Iranian Children Aged 2–5 Years: A National Cross-Sectional Analysis

**DOI:** 10.3390/nu17101631

**Published:** 2025-05-09

**Authors:** Maryam Sadat Kasaii, Sara Rodrigues, Morteza Abdollahi, Anahita Houshiar-Rad, Julian Perelman

**Affiliations:** 1Faculty of Nutrition and Food Sciences, University of Porto, 4150-180 Porto, Portugal; saraspr@fcna.up.pt; 2Social Determinants of Health Research Center, Shahid Beheshti University of Medical Sciences, Tehran 1546817613, Iran; morabd@yahoo.com; 3Department of Nutrition Research, National Nutrition and Food Technology Research Institute (WHO Collaborating Center), Faculty of Nutrition and Food Technology, Shahid Beheshti University of Medical Sciences, Tehran 1981619573, Iran; anahrad@yahoo.com; 4Nova National School of Public Health and Comprehensive Health Research Center, Nova University of Lisbon, 1600-407 Lisbon, Portugal

**Keywords:** socioeconomic, SES, stunting, overweight, malnutrition

## Abstract

**Background/Objectives:** Evidence indicates a high prevalence of stunting and overweight among Iranian children. This study explores their socioeconomic patterning and the mediating role of nutrition adequacy. **Methods:** The data were derived from the most recent 2017 Demography and Health Survey and the Multiple Indicator Cluster Survey, which were conducted in Iran. Children aged between 2 and 5 years were selected for the study through a two-stage random sampling process (*n* = 11,147). The probability of stunting and overweight was modeled using logistic regression. Parental education, occupation, and living conditions (areas, rooms, and assets of the household) were explanatory variables, with the diet diversity score (DDS) as a mediator. Analyses were adjusted for age and sex. **Results:** Children had over 1.7 times higher odds of stunting with a primary-educated father [95% CI: 1.13–2.62] and twice the odds with an illiterate mother [95% CI: 1.30–3.30]. The risk of stunting was almost 1.5 higher in children living in smaller houses [95% CI: 1.12–2.04]. Finally, a significant association was observed between low asset ownership and stunting [OR = 2.01; 95% CI: 1.23–3.27]. The results showed no significant relationship between socioeconomic factors and children’s overweight, indicating that overweight was less socially patterned. Higher DDS was associated with lower stunting and higher overweight prevalence but did not mediate the effects of socioeconomic status. **Conclusions**: Stunting disproportionately affects children from households with a lower socioeconomic background in Iran. Parental education, area, and assets were key factors, highlighting the need for targeted nutrition education programs.

## 1. Introduction

Malnutrition is one of the major problems in low and middle-income countries and can lead to stunting, underweight, and overweight. Indeed, malnutrition affects the growth, development, and health of children under 5 years of age; more than 45% of deaths in children under five years of age in low and middle-income countries are related to undernutrition [1]. Stunting in early childhood has both immediate and long-term consequences, including higher risk of illness, impaired cognitive and physical development, and reduced learning capacity [2]. Over time, it can lead to non-communicable diseases such as diabetes and hypertension, lower work productivity, and adverse reproductive outcomes [3]. Children who gain excess weight after age two are also more likely to develop obesity later in life, increasing the burden of nutrition-related health issues in adulthood [4]. According to World Health Organization (WHO) statistics, although the prevalence of short stature decreased by 11% from 2000 to 2020, 149 million children under the age of 5 are still affected by stunting, and the rate of overweight in children increased by 0.3% during this period [5]. In 2019, the prevalence of overweight among Iranian children under the age of five was reported 9% [6]. Between 2010 and 2018, the prevalence of stunting among Iranian children under 5 years old increased from 10 to 12% [7,8].

The literature has linked malnutrition and inadequate dietary patterns to low socioeconomic status (SES). SES has been proxied by various indices like occupation, education, income, wealth level, assets, or a combination of them [9,10]. A study that investigated the trends in socioeconomic inequality in malnutrition among children under 59 months of age in Nigeria showed that the rate of stunting in children decreased with increasing parental education and wealth [11]. An investigation in Bangladesh revealed that children who were stunted had a lower diet diversity score and low economic and social status, while mothers’ low literacy levels were associated with short stature [12]. In another study conducted on Indian children, the rate of stunting was 40%, and stunting was more prevalent among children from illiterate mothers and among girls and children from lower-class areas [13]. In a study conducted in Vietnam in 2016, the findings showed the relationships between the region of residence, race, mother’s literacy level, economic and social class, and children’s stunting. As the education level of mothers increases, the rate of short stature in children decreases significantly [14].

In Iran, in 2010, a study was conducted on children under five years old, which showed that stunting was much more common than wasting or underweight. Short stature was significantly greater in rural areas and in lower socioeconomic categories [7]. These studies are repeated every few years to monitor children’s health and inform national policies.

The aim of this study was to update these results with more recent data, extend its objective to a larger array of outcomes and socioeconomic measures, and assess the mediating role of diet. Given the limited evidence on the mediating effects of dietary diversity in the Iranian context, this research provides a novel contribution to the literature. By filling this gap, the study provides a better understanding on the pathway between socioeconomic variables and stunting, aiming at providing deeper insights that can support more targeted and effective nutrition interventions.

## 2. Materials and Methods

### 2.1. Sample

The data for this cross-sectional study were sourced from the “Child Health Aspect: Anthropology, Development, Nutrition” survey, which was conducted in 2017 across 31 Iranian provinces [15]. The sampling method involved random selection of households with children under five years old using the SIB, NAB, and SINA software systems. These systems contain the names and addresses of households registered in the Primary Healthcare (PHC) network. They served as the sampling frame, ensuring that only households with active and up-to-date records in the PHC system were included. A total of 18,600 households with children under the age of 5 were randomly selected using a two-stage sampling system (provinces and households). After invitation to the healthcare center, the mother or tutor of the child was surveyed by a nutritionist using a face-to-face interview. The questionnaire used was based on the DHS 2010 (Demographic and Health Survey, WHO) and MICS6 (Multiple Indicator Cluster Survey, UNICEF) questionnaires [16,17], and the data were gathered through self-reported responses. The participants were asked about their general household information (location, area of residence, marital status of the caregiver, sex of the caregiver, number of household members, and number of children under 5 years old), child information (age, sex, birth weight, child diseases in the past 15 days, and child birth order), type of child nutrition and food consumed in the last 24 h, child growth rate, and demographic and socioeconomic information about the parents. In addition, the child’s weight was measured with an electronic UNISCALE scale with an accuracy of 0.1 kg, and the child’s height was measured with a stadiometer with an accuracy of 0.1 cm. According to the surveys, 1 in 4 children under the age of 5 was below the standard child growth chart of the WHO [18]. In the present study, the aim was to explore the effect of dietary diversity on stunting and overweight in children under 5 years of age in conjunction with socioeconomic indicators. Considering that children under 2 years of age primarily grow through breastfeeding and complementary feeding, measuring dietary diversity in this age group is not particularly pertinent. As a result, the focus was on children aged 24–59 months, resulting in a final sample of 11,147 observations. Missing data were handled using listwise deletion; cases with missing data in key variables were excluded from the analysis.

The database used in the present paper came from a survey that was approved by the Research Council of National Nutrition and Food Technology Research Institute and Ethics Committee of National Nutrition and Food Technology Research Institute (reference code: IR.SBMU.NNFTRI.REC.1396.165). Prior to participating in the survey, written informed consent was obtained from all mothers or tutors of children. The datasets used during the current study are available from MA on reasonable request.

### 2.2. Outcome

The WHO mentions that malnutrition may take various forms, such as stunting, underweight, wasting, and overweight [18]. Although stunting is one of the obvious consequences of growth failure in children [19], the prevalence of childhood obesity is increasing worldwide [20]. Stunting and overweight were identified as the most challenging variables, prompting the decision to concentrate on these two aspects [21,22]. According to the Centers for Disease Control and Prevention (CDC), stunting is defined as a height-for-age z score (HAZ) less than −2 standard deviations (SDs) below the mean according to the WHO child growth standard [20]. Stunting was coded as a binary variable. According to the CDC, overweight is defined as a weight-for-height z score (WHZ) above 2 (+2) SDs above the mean according to the WHO child growth standard. The overweight variable was also coded as binary.

### 2.3. Explanatory Variables

The following socioeconomic variables related to stunting and/or overweight in the literature were considered in the analysis: parental education [23,24], parental occupation [9], and household assets as proxies for wealth [11,25,26]. The father’s and mother’s education were coded into five categories: illiterate, primary school, secondary school, high school or diploma, and university education. The occupations of the fathers were classified into 9 groups according to employment status: employee, worker, farmer, shopkeeper, retired, student, unemployed, military, and others. Some occupations were grouped in the study because of the expected income similarity and the small number of people within some categories; namely, retired, unemployed, and students were grouped into a single class. Due to the high proportion of housewives and the small number of mothers across other occupations, maternal employment status was categorized into two groups, namely, working and nonworking mothers [27]. Employees, workers, farmers, and shopkeepers were classified as working mothers, and retirees, students, unemployed individuals, and housewives were considered nonworking mothers. This binary classification aimed to facilitate statistical analysis and ensure adequate sample sizes within each group.

Regarding assets, the household’s number of rooms, the house area, and a list of available goods were used in the analysis. The number of rooms in each household was divided by its size, resulting in the categorization of the variable into four quartiles. The household’s house area was divided by the household number of members, and the variable was divided into quintiles. The household assets were coded as the sum of the number of assets, from zero to five. The following assets were considered: TV, freezer, microwave, computer, and car.

### 2.4. Mediation Variable

The diet diversity score (DDS) is an index to measure variety of the dietary items consumed in the day [28]. The DDS was used to assess nutrient adequacy [29]. The DDS was calculated by counting the number of food groups that every child consumed on the last day. Mothers were asked qualitative, self-report questions about their children’s dietary habits, for example, “Has your child consumed any of the following foods in the past 24 h?”. Food items were classified into 7 groups: cereals, fruits and vegetables containing vitamin A, other fruits and vegetables, legumes, eggs, flesh foods, and dairy products. The minimum score was one, and the maximum score was seven.

According to the children’s ages, almost all the children had consumed 3 food groups—cereals, fruits, and legumes—in the last 24 h, and the number of children who consumed less than that was rare. In this study, the quality of the diet was therefore calculated based on the consumption or lack of consumption of the remaining 4 food groups (eggs, flesh foods, dairy products, and fruits and vegetables containing vitamin A) by the child in the last 24 h. If they consumed all 4 food groups, they were included in the high-quality group; if they ate fewer than these 4 food groups, they were assigned to the low-quality group. This modified approach was chosen because it better reflects the variability in food group consumption within the study population.

### 2.5. Statistical Analysis

Univariate logistic regression was initially conducted for each explanatory variable independently. These included child’s age and sex, parental education and occupation, and household wealth indicators (such as house area, number of rooms, and household assets). Subsequently, multivariate logistic regression was performed including all variables to identify the most relevant socioeconomic factors associated with stunting and overweight while adjusting for age and sex. The results of multilevel logistic regression are presented as odds ratios (ORs) with 95% confidence intervals (95% CIs), and the significance level was defined as a *p*-value lower than 0.05. In the final model, logistic regression was subsequently performed including the DDS to measure the mediation role of the diet diversity score in the SES–nutrition relationship. The data analysis was performed with SPSS software version 21.

## 3. Results

### 3.1. Sample Description and Univariate Statistics

Of the 11,147 children included in the study, 50.9% were male (Table 1). In terms of age, 36.9% of the children were between 2 and 3 years old. A total of 52.7% of the fathers had a high school education or above, and 3.4% of them were illiterate. A total of 4.6% of mothers had no education, and 55.6% of mothers had at least a high school education. The most common job among fathers was shopkeeping (41%). A total of 3.8% of fathers were unemployed, and 91.4% of mothers did not work. Approximately two-thirds of the households had two or fewer rooms (63.4%), and 12.4% of the households had no assets (Table 2).

Stunting: The nationally weighted prevalence of stunting among children aged 2–5 years was 5.1%, adjusted for age, gender, and area of residence using weights derived from the 2016 national census (Table 2). According to the fathers’ education levels, 12.2% of the illiterate fathers had stunted children, whereas this percentage was 2.7% among fathers with the highest level of education [OR = 5.21; 95% CI: 3.49–7.79]. In addition, there was a distinct gradient where the risk of stunting decreased as paternal literacy increased from illiterate to primary, secondary, and high school levels [OR = 5.21, 3.62, 1.93, 1, 49]. Children’s short height was observed in 14.4% of the illiterate mothers, compared to 2.8% of those who had attained the highest level of education [OR = 5.94; 95% CI: 4.17–8.45]. The highest percentage of children with short stature was observed for retired, student, or unemployed fathers compared to employed (9.6 versus 2.4%, OR = 4.41). Moreover, stunting was 4.4, 2.95, 2.83, and 1.88 times more prevalent among children whose fathers were nonworking (retired, student, or unemployed), farmers, laborers, or shopkeepers, respectively (Figure 1).

Stunting was 1.8 times more common in households with the lowest number of rooms per person [OR = 1.89; 95% CI: 1.45–2.48]. Regarding living area, 8.4% of children living in smaller areas were stunted, whereas 3.5% of children living in larger homes were stunted [OR = 2.52, 95% CI: 1.95–3.25]. A total of 11.2% of households without assets have short children, while 2.4% of households with five assets have short children. Households possessing fewer than three assets had a higher likelihood of having stunted children compared to those with five assets. A higher diet diversity score was related to a lower risk of stunting among children [OR = 0.60, 95% CI: 0.53–0.69] (Table 2).

Overweight: The prevalence of overweight in children aged 2 to 5 years was 2.4%. Children whose parents had a primary education were 44% less likely to be overweight than were those whose parents had a higher diploma [OR = 0.56; 95% CI: 0.37–0.85]. In terms of occupation, employees (2.8%), shopkeepers (2.9%), and soldiers (3%) had the highest number of overweight children. In contrast, the children of working fathers, farmers, and unemployed fathers were less prone to overweight. A smaller home area was associated with a reduced risk of overweight [OR = 0.63; 95% CI: 0.44–0.91]. Households with zero or one asset were significantly less overweight than households with five assets [OR = 0.39; OR = 0.60]. The prevalence of overweight was much greater in children who ate all seven food groups [OR = 1.39, 95% CI: 1.08–1.81] (Table 3).

### 3.2. Multivariate Regression

Stunting: The significant associations between stunting and socioeconomic variables remained after all variables were included but only for those from the lowest socioeconomic groups (i.e., there was no longer a clear gradient). Stunting was 70% and 73% more common among children who had illiterate and primary-educated fathers, respectively, than among children with a father with a diploma or higher education [OR = 1.70; 95% CI: 1.01–2.88] [OR = 1.73; 95% CI: 1.14–2.64]. Illiterate mothers had twice as many short children as literate mothers [OR = 2.07; 95% CI: 1.30–3.29]. Small house area and lack of home assets were directly related to short stature [OR = 1.51; 95% CI: 1.12–2.04] [OR = 2.06; 95% CI: 1.27–3.35] (Table 2).

Overweight: In contrast, regarding overweight status, only nonworking fathers were less likely to have overweight children [OR = 0.29; 95% CI: 0.09–0.99], while associations with other variables were no longer significant after all variables were included (Table 3).

### 3.3. Mediation Analysis

Stunting and overweight: First, after adding the DDS to the regression, most socioeconomic variables that were significant remained so even if their values slightly decreased, except for the father’s illiteracy, which was no longer significant. Second, when assessing the combined impact of socioeconomic status and the DDS, no significant relationship was observed between short stature or overweight and the DDS (OR = 0.92; 95% CI: 0.74–1.13; OR = 1.29; 95% CI: 0.99–1.69). Finally, the associations with father retirement, student status, and unemployed status were no longer significant for either stunted or overweight individuals (OR = 1.26; 95% CI: 0.42–3.78; OR = 1.44; 95% CI: 0.79–2.62) (Table 2 and Table 3).

## 4. Discussion

The probability of having a stunted child among illiterate parents was greater than among educated parents. The mother’s occupation was not significantly correlated with short stature or overweight. While univariate analysis showed that living in a smaller house was associated with a higher prevalence of stunting, the number of rooms was not independently linked to child height in the multivariate model after adjusting for other socioeconomic factors. However, having fewer assets had a consistent relationship with the children stunting. None of the housing conditions were related to the overweight status of the children. Finally, the diet diversity score did not explain the relationship between SES and stunting or overweight among children.

### 4.1. Stunting

#### 4.1.1. Parental Education

These findings align with previous research, including a MICS study in Bangladesh, which found that children with more educated fathers were less likely to be stunted [30]. Furthermore, another study of children in Rishikesh showed that stunting was more prevalent in children whose fathers were illiterate [31]. In contrast, Mutiarasari et al. (2021) reported that there was no direct link between paternal education and child stunting. Instead, the greater influence of maternal education and awareness on child growth was emphasized, as mothers are typically more involved in caregiving [32]. These findings support this perspective and are consistent with similar studies conducted in in Indonesia, Rwanda, Ethiopia, and sub-Saharan Africa [33,34,35,36,37]. Although a previous study showed that there was no relationship between stunting and maternal education level in Pakistan [38], some surveys acknowledged that educated parents can decrease the possibility of stunting in children [30]. These findings suggest that education plays an important role in preventing stunting [39]. A higher level of literacy may lead parents to learn more about care and the nutritional status of children under five; therefore, they provide diets with more necessary nutrients for height growth [32].

#### 4.1.2. Parental Occupation

Notably, our findings indicated no correlation between parental occupation and child stunting. This aligns with the findings of an Indonesian study that asserted the lack of a connection between the mother’s occupation and child stunting [40]. In our sample, a large majority of women were not employed (91%), leading to low variability in working conditions, which may explain the lack of association. In addition, according to a 2021 study, only 12.7% of households in Iran are female-headed [41]. This may explain why the occupation of mothers has not played a significant role in families’ living expenses in Iran. Our findings are inconsistent with those of studies conducted in Sub-Saharan Africa, which reported no significant relationship between the level of education and the profession of the household head and stunting [42]. While the research of Chowdhury et al. showed that children under five whose fathers do not work are more likely to have multiple concurrent forms of undernutrition than children whose fathers are businessmen [43], our study did not find such an association. Some studies have linked fathers’ unemployment to lower income and limited access to food and healthcare, increasing the risk of stunting [40,44]. However, this effect may be better captured through household wealth indicators rather than occupation alone. For instance, in Bangladesh, Nahar et al. found that children living in rural areas were more likely to experience stunting than those in urban settings, likely due to disparities in access to healthcare, education, and nutrition services [45]. This study contradicts two studies in Bangladesh and India that investigated the factors affecting the height of children under 5 years of age and showed that mothers under 18 years of age and working are more likely to have stunted children [46,47]. In another study, mothers who worked 25 to 40 h per week were more likely to have a short child than mothers who did not work [48]. Some studies believe that, although occupied parents might have greater nutritional knowledge, they have limited time to allocate to preparing their children’s meals. Thus, parents entrust their children to their families and caregivers, but there is a weak control over their children’s diets. For this reason, several studies have shown that stunting is more prevalent in children with working parents [49]. While these findings suggest a possible link between parental employment and child stunting in some settings, our results indicate that occupation alone is not a determining factor in the Iranian context.

#### 4.1.3. Housing and Assets

In terms of housing and assets, the prevalence of stunting increased as SES decreased. In 2019, Asuman et al. conducted a study with demographic and health survey data from 10 African countries and reported that as the SES of households decreased, the prevalence of stunting increased accordingly [50]. This result was consistent with similar studies in Peru, Indonesia, Egypt, India, Ethiopia, and Bangladesh [21,36,37,47,50,51,52]. According to the MICS 2019 study in Bangladesh, children from richer families had a 51% lower chance of being stunted [30]. In addition, in the study of Papua New Guinea, children from households with lower wealth quintiles were shown to be more likely to be stunted than children from households with higher wealth quintiles [24]. In general, having more assets means an increase in purchasing power and, as a result, leads to improved access to nutritious foods and healthcare for the child. A child with a lower socioeconomic level is less likely to have an adequate food intake and, as a result, tends to be more malnourished. Therefore, a higher level of wealth and assets may prevent children from stunting [36]. In an Indian study, it is stated that children who lived in wooden houses (without concrete) were shorter than children who lived in cement houses [46]. On the other hand, some studies announced that there is no relationship between SES and stunting [32,38,53]. However, several studies show that a considerable proportion of stunted children are from low-income households [32,50].

#### 4.1.4. Diet Quality

Considering that stunting is one of the long-term signs of malnutrition that affects children’s health and can affect adult health, special emphasis should be placed on recognizing the crucial role of nutrition in preventing stunting [42]. In this study, the DDS was not a significant factor affecting the relationship between SES and stunting. These results are in line with the findings of Gassara et al. and Azupogo et al. that indicated that the DDS was not related to stunting in children under 59 months who lived in Chad and Ghana [42,54]. In contrast, another study in Myanmar found that children who consumed fewer than four food groups were more exposed to stunting [55]. Rah et al. in Bangladesh reported that as the DDS decreased, the number of stunted children increased [12]. Many studies, including systematic reviews, have been conducted in sub-Saharan Africa, and research in Cambodia, Western China, and India has shown that dietary diversity scores influence malnutrition, especially stunting [56]. However, some studies also acknowledged that the DDS is not the only factor affecting stunting. Other factors not included in the present study, such as genetics, infectious diseases, and food and nutrient intake quantities, can also affect stature [57,58]. This could explain why SES differences in stunting were not fully mediated by the DDS in our study. While previous research has found diet diversity to mediate the SES–malnutrition relationship, our null mediation finding may reflect the limited variability in the DDS across SES groups in our sample. Additionally, broader structural factors—such as food access and cultural dietary norms—may have a stronger influence on children’s diets, weakening the mediating effect of the DDS. These results underscore the complexity of malnutrition pathways and highlight the need for further research exploring additional behavioral and environmental mediators.

### 4.2. Overweight

Although no statistically significant associations between SES and overweight were found, trends such as higher overweight prevalence among children from lower-SES households—specifically those with fewer assets or smaller living spaces—were observed. These findings suggest that socioeconomic disparities may exist but were not detected statistically, possibly due to sample size limitations. Regional studies have also reported mixed results, indicating that these relationships may vary depending on the social and environmental context. In contrast, a study conducted in China claimed that housewife mothers may have more obese children than mothers who work more than 40 h [48]. In another study of 18 children in sub-Saharan Africa, the results showed that the probability of overweight children is greater for working mothers [34]. However, our findings are consistent with another study showing no relationship between the wealth index and overweight in children. In Pakistan, children living in the richest households were less likely to be overweight [59]. On the contrary, Ekholuenetale et al., who studied children under 59 months of age in 35 sub-Saharan African countries, found that overweight was observed in households with the highest wealth scores. In addition, this research emphasized that overweight in children is significantly related to mothers’ education, and overweight children were more common among mothers with lower education levels [60].

In line with our results, Diallo et al. showed that none of the parameters, namely, maternal education, gender of children, maternal occupation, or socioeconomic status of households, were related to overweight in children younger than five years [61]. However, in general, people with low socioeconomic indices in industrialized countries and population groups with high socioeconomic indices in developing countries are more likely to be obese [62]. In the present study, no significant relationship was observed between overweight and the diet diversity score. Conversely, in another study in Ghana, there was a positive association between the DDS and weight-for-height Z-score among 24- to 59-month-old children [54]. Other findings from a study of Tunisian children revealed that an elevated diet diversity score correlated with an increased prevalence of overweight in children. The expansion of dietary variety likely results in increased energy intake, leading to increased fat consumption and an elevated risk of overweight [63]. Based on the study of Lioret et al., overweight in children has a stronger relationship with the education level of parents compared to other socioeconomic factors [64]. The absence of significant associations between socioeconomic factors and child overweight in Iranian children may be explained by the relatively low prevalence of overweight in this group (2.4%). Additionally, various factors such as genetics, cultural norms, age, and access to healthcare could influence these outcomes, suggesting that socioeconomic factors alone may not fully capture the complexities behind childhood overweight in Iran.

### 4.3. Limitations and Strengths of the Study

This study was derived from a national representative Iranian survey that representatively describes the anthropometric status of children and its relationship with socioeconomic indicators. As a result, the nutritional status of one of the important age groups in Iranian society is represented. Moreover, the other strengths of this study are its accurate and large sample size. However, as the sample size increases, collecting additional indicators becomes more challenging for field work. For instance, this study involved only questions about children’s qualitative food consumption data from the past 24 h. Therefore, while this study assessed food quality qualitatively, it is recommended that future research incorporate inquiries into food quantity (such as a 24 h recall). This approach enables a comprehensive examination of children’s macronutrient and micronutrient intake, providing insights into the presence of deficiencies, which constitute another subset of malnutrition according to WHO guidelines. Another limitation of this study was the self-reporting of questions about the SES of the household, which may lead to overestimation due to social desirability bias, as well as recall bias affecting the accuracy of responses [65]. Although household assets, room area, and number of rooms were used as proxies for socioeconomic status, a direct measure of household income was preferable but not available in the dataset. In addition, it was not possible to measure how the association between SES and stunting varied across ages or regions. Although the narrow range of ages included would imply minimal differences, the variety of economic and lifestyle conditions among Iranian regions could affect the explored associations. For instance, a study declared that malnutrition was more common among children in rural areas than those who live in urban areas [66]. Given the cross-sectional nature of this study, causality cannot be inferred between socioeconomic factors, dietary diversity, and child nutritional outcomes. To measure causal relationships, longitudinal data should be used, which are not available for Iran.

## 5. Conclusions

This article investigated the effect of socioeconomic indicators along with diet diversity on two common types of malnutrition, stunting and overweight. In this study, the education of parents, house area, and house assets were found to be protective factors against stunting among 24–59-month-old children. Although no significant associations between SES and overweight were found, trends like lower overweight prevalence in households with fewer assets or smaller living spaces suggest emerging disparities. To reduce these socioeconomic inequalities, intervention strategies should target vulnerable families, specifically those with parents with lower educational attainment, retired and unemployed fathers, and limited house area and few assets. These interventions could include conditional cash transfer programs, housing support, parental education initiatives focused on nutrition and caregiving, and community-based child nutrition services such as daycare or preschool feeding programs [67].

## Figures and Tables

**Figure 1 nutrients-17-01631-f001:**
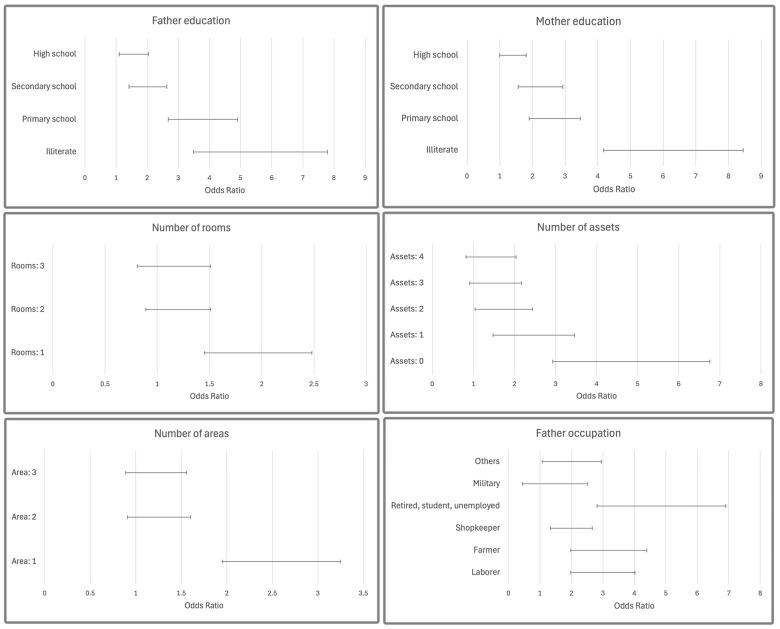
Association between parents’ socioeconomic status (SES) and children’s stunting: odds ratios with 95% confidence intervals.

**Table 1 nutrients-17-01631-t001:** Distribution of age and sex among Iranian children—Children Anthropometry, Nutrition and Development Survey, 2017.

	Age (Month)	24–35	36–47	48–59	Total
Sex *n* (%)	Female	1970 (49.2)	1773 (48.1)	1732 (50.0)	5475 (49.1)
	Male	2033 (50.7)	1909 (51.8)	1730 (49.9)	5672 (50.9)

**Table 2 nutrients-17-01631-t002:** Association of parents’ socioeconomic status (SES), diet diversity score (DDS), and children’s stunting—Children Anthropometry, Nutrition and Development Survey, 2017.

		Stunting (HAZ) < −2			
Characteristics		Total *n* (%)	Stunting *n* (%)	Adjusted OR without Diet Quality (95% CI)	Adjusted OR with Diet Quality (95% CI)
Education (father)	Illiterate	387 (3.4)	47 (12.2)	1.70 * (1.01–2.88)	1.69 (0.99–2.86)
	Primary school	1976 (17.7)	174 (8.8)	1.73 * (1.14–2.64)	1.72 * (1.13–2.62)
	Secondary school	2885 (25.9)	142 (4.9)	1.16 (0.77–1.74)	1.16 (0.77–1.74)
	High school	3690 (33.1)	144 (3.9)	1.09 (0.75–1.59)	1.08 (0.74–1.58)
	Higher education	2190 (19.6)	58 (2.7)	Ref	Ref
Education (mother)	Illiterate	511 (4.6)	74 (14.4)	2.07 * (1.30–3.29)	2.07 * (1.30–3.30)
	Primary school	2326 (21.0)	158 (6.8)	1.21 (0.82–1.79)	1.21 (0.82–1.79)
	Secondary school	2049 (18.5)	119 (5.8)	1.32 (0.90–1.93)	1.32 (0.91–1.93)
	High school	3974 (35.9)	149 (3.8)	1.02 (0.72–1.43)	1.02 (0.72–1.43)
	Higher education	2180 (19.7)	62 (2.8)	Ref	Ref
Occupation (father)	Employee	1572 (14.2)	38 (2.4)	Ref	Ref
	Laborer	2604 (23.5)	171 (6.6)	1.42 (0.92–2.21)	1.22 (0.79–1.87)
	Farmer	1016 (9.1)	70 (6.9)	1.35 (0.83–2.20)	1.16 (0.72–1.87)
	Shopkeeper	4535 (41.0)	205 (4.5)	1.36 (0.90–2.06)	1.17 (0.76–1.78)
	Retired, student, or unemployed	423 (3.8)	41 (9.6)	1.69 (1.00–2.87)	1.26 (0.42–3.78)
	Military	232 (2.1)	6 (2.6)	1.10 (0.46–2.66)	0.94 (0.37–2.37)
	Others	664 (6.0)	30 (4.5)	1.17 (0.67–2.03)	0.85 (0.49–1.48)
Occupation (mother)	Working ^1^	952 (8.5)	34 (3.5)	Ref	Ref
	Nonworking	10,168 (91.4)	528 (5.1)	0.98 (0.69–1.39)	1.03 (0.72–1.45)
Rooms/family size (quartiles)	1	2489 (22.5)	179 (7.2)	1.06 (0.79–1.43)	1.06 (0.78–1.43)
	2	4508 (40.9)	204 (4.5)	0.93 (0.71–1.23)	0.93 (0.71–1.23)
	3	1967 (17.8)	86 (4.4)	1.07 (0.78–1.47)	1.06 (0.77–1.45)
	4	2054 (18.6)	81 (4.0)	Ref	Ref
Areas/family size (quartiles)	1	2742 (24.6)	231 (8.4)	1.51 * (1.12–2.04)	1.50 * (1.11–2.03)
	2	2868 (25.7)	120 (4.2)	0.91 (0.66–1.25)	0.90 (0.66–1.24)
	3	3134 (28.1)	129 (4.1)	1.08 (0.80–1.45)	1.08 (0.80–1.45)
	4	2382 (21.4)	84 (3.5)	Ref	Ref
Assets (score = sum of assets)	0	1393 (12.4)	156 (11.2)	2.06 * (1.27–3.35)	2.01 * (1.23–3.27)
	1	1939 (17.3)	117 (6.0)	1.35 (0.84–2.17)	1.33 (0.83–2.15)
	2	2479 (22.2)	105 (4.2)	1.14 (0.72–1.83)	1.13 (0.71–1.81)
	3	2294 (20.5)	86 (3.7)	1.14 (0.71–1.82)	1.13 (0.71–1.80)
	4	1818 (16.3)	70 (3.8)	1.24 (0.77–2.00)	1.23 (0.77–1.98)
	5	983 (8.8)	24 (2.4)	Ref	Ref
Diet Diversity Score (DDS)		11,127	563 (5.1)		0.92 (0.74–1.13)

Note: ^1^ Working mothers (employees, laborers, farmers, shopkeepers, and others); nonworking mothers (retired, students, unemployed, and housework); * significance at a *p* value < 0.05, OR: odds ratio, SES: socioeconomic status.

**Table 3 nutrients-17-01631-t003:** Association of parents’ socioeconomic status (SES), diet diversity score (DDS) and children’s overweight—Children Anthropomerty, Nutrition and Development Survey, 2017.

			Overweight (WHZ) > +2			
Characteristics		Total *n* (%)	Overweight *n* (%)	Unadjusted OR (95% CI)	Adjusted OR (95% CI)	Adjusted OR with Diet Quality (95% CI)
Education (father)	Illiterate	387 (3.4)	12 (3.1)	1.05 (0.56–1.96)	2.05 (0.92–4.58)	2.18 (0.97–4.87)
	Primary school	1976 (17.7)	33 (1.7)	0.56 * (0.37–0.85)	0.83 (0.47–1.46)	0.85 (0.48–1.50)
	Secondary school	2885 (25.9)	49 (1.7)	0.57 * (0.39–0.83)	0.67 (0.41–1.10)	0.69 (0.42–1.13)
	High school	3690 (33.1)	108 (2.9)	1.01 (0.74–1.38)	1.12 (0.75–1.66)	1.14 (0.77–1.70)
	Higher education	2190 (19.6)	64 (2.9)	Ref	Ref	Ref
Education (mother)	Illiterate	511 (4.6)	11 (2.2)	0.75 (0.39–1.44)	1.37 (0.60–3.09)	1.36 (0.60–3.07)
	Primary school	2326 (21.0)	37 (1.6)	0.55 * (0.37–0.83)	0.95 (0.56–1.62)	0.96 (0.56–1.63)
	Secondary school	2049 (18.5)	46 (2.2)	0.79 (0.54–1.16)	1.24 (0.77–1.98)	1.24 (0.77–1.98)
	High school	3974 (35.9)	109 (2.8)	0.98 (0.72–1.35)	1.29 (0.89–1.86)	1.28 (0.88–1.86)
	Higher education	2180 (19.7)	61 (2.8)	Ref	Ref	Ref
Occupation (father)	Employee	1572 (14.2)	44 (2.8)	Ref	Ref	Ref
	Laborer	2604 (23.5)	48 (1.9)	0.66 (0.44–1.00)	0.80 (0.48–1.33)	1.25 (0.64–2.47)
	Farmer	1016 (9.1)	16 (1.6)	0.56 * (0.31–0.99)	0.71 (0.37–1.37)	0.97 (0.51–1.87)
	Shopkeeper	4535 (41.0)	132 (2.9)	1.05 (0.74–1.48)	1.17 (0.77–1.77)	0.87 (0.40–1.87)
	Retired, student, or unemployed	423 (3.8)	2 (2.3)	0.25 * (0.08–0.80)	0.29 * (0.09–0.99)	1.44 (0.79–2.62)
	Military	232 (2.1)	7 (3.0)	1.10 (0.49–2.46)	1.08 (0.48–2.45)	1.12 (0.24–5.14)
	Others	664 (6.0)	13 (2.0)	0.70 (0.37–1.33)	0.82 (0.41–1.61)	0.15 (0.02–1.20)
Occupation (mother)	Working ^1^	952 (8.5)	32 (3.3)	Ref	Ref	Ref
	Nonworking	10,168 (91.4)	232 (2.2)	1.37 (0.72–1.86)	0.75 (0.50–1.13)	1.32 (0.88–1.97)
Rooms/family size (quartiles)	1	2489 (22.5)	57 (2.3)	0.72 (0.50–1.03)	1.10 (0.73–1.67)	1.12 (0.74–1.70)
	2	4508 (40.9)	89 (2.0)	0.62 * (0.45–0.86)	0.77 (0.55–01.09)	0.78 (0.55–1.10)
	3	1967 (17.8)	50 (2.5)	0.81 (0.56–1.17)	0.86 (0.58–1.26)	0.86 (0.58–1.25)
	4	2054 (18.6)	65 (3.2)	Ref	Ref	Ref
Areas/family size (quartiles)	1	2742 (24.6)	53 (1.9)	0.63 * (0.44–0.91)	0.75 (0.50–1.14)	0.75 (0.50–1.14)
	2	2868 (25.7)	55 (1.9)	0.62 * (0.44–0.89)	0.71 (0.47–1.05)	0.70 (0.47–1.04)
	3	3134 (28.1)	87 (2.8)	0.92 (0.67–1.26)	0.98 (0.70–1.37)	0.98 (0.70–1.36)
	4	2382 (21.4)	71 (3.0)	Ref	Ref	Ref
Assets (score = sum of assets)	0	1393 (12.4)	19 (1.4)	0.39 * (0.22–0.69)	0.62 (0.32–1.20)	0.66 (0.34–1.28)
	1	1939 (17.3)	35 (1.8)	0.60 * (0.38–0.96)	0.90 (0.53–1.52)	0.93 (0.55–1.58)
	2	2479 (22.2)	54 (2.2)	0.67 (0.44–1.04)	0.88 (0.54–1.42)	0.90 (0.56–1.47)
	3	2294 (20.5)	72 (3.1)	0.89 (0.58–1.35)	1.05 (0.66–1.65)	1.07 (0.68–1.69)
	4	1818 (16.3)	48 (2.6)	0.75 (0.48–1.17)	0.84 (0.52–1.35)	0.85 (0.53–1.37)
	5	983 (8.8)	26 (2.7)	Ref	Ref	Ref
Diet Diversity Score (DDS)		11,127	265 (2.4)	1.39 * (1.08–1.81)		1.29 (0.99–1.69)

Note: ^1^ Working mothers (employees, laborers, farmers, shopkeepers, and others); nonworking mothers (retired, students, unemployed, and housework); * Significance at a *p* value < 0.05, OR: odds ratio, SES: socioeconomic status.

## Data Availability

The data presented in this study are available on request from M.A. due to privacy and ethical restrictions.

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
