# Peer review of "Socioeconomic Patterning of Stunting and Overweight Among Iranian Children Aged 2–5 Years: A National Cross-Sectional Analysis"

_nutrients, 2025, doi:10.3390/nu17101631_

Round 1

Reviewer 1 Report

Comments and Suggestions for Authors

This study represents an interesting research question that is relatively understudied in the literature. There are a number of limitations with the methods (such as the measure of diet quality) but the authors do a reasonable job of discussing them in the discussion.

Some specific recommendations are below:

Lines 68-74: I don’t think this paragraph should be here.

Line 78: What are SIB, NAB, and SINA?

Line 156: This sentence states “…included all the variables in the model”. Please state the specific variables in the model and whether they explained a significant amount of the variance. It appears to be age and sex only (line 160), but I think a clarification is needed. It is confusing to understand the what the “adjusted OR without diet quality” means in the tables.

Line 239: “illitrcy” appears to be a spelling error.

Line 345: “overweightin” should be separated.

Lines 394-395: The authors state “…were found to be effective at preventing stunting…”. These results are cross-sectional in nature, therefore causation cannot be determined. The word “preventing" implies causation and shouldn’t be used.

Author Response

Comment 1: Lines 68-74: I don’t think this paragraph should be here.

Response 1: You are right. Thank you for your consideration. I deleted that.

Comment 2: Line 78: What are SIB, NAB, and SINA?

Response 2: We agree with this comment therfore, I added in Line 80-83.

Comment 3: [Line 156: This sentence states “…included all the variables in the model”. Please state the specific variables in the model and whether they explained a significant amount of the variance. It appears to be age and sex only (line 160), but I think a clarification is needed.] It is confusing to understand the what the “adjusted OR without diet quality” means in the tables.

Response 3: We changed it in line 170-174

Comment 4: [Line 239: “illitrcy” appears to be a spelling error.]

Response 4: Thank you for your attention. We fixed it in line 250 now.

Comment 5: [Line 345: “overweightin” should be separated.]

Response 5: We agree with this comment therfore, I fixed it in line 357.

Comment 6: [Lines 394-395: The authors state “…were found to be effective at preventing stunting…”. These results are cross-sectional in nature, therefore causation cannot be determined. The word “preventing" implies causation and shouldn’t be used.]

Response 6: We agree with this comment therfore, I changed it to protective factor in line 411.

Reviewer 2 Report

Comments and Suggestions for Authors

This is an interesting research study with quite adequate novelty. However, some points should be addressed.

  • In general, the Introduction section is quite short and needs to be enriched by additional data in order to emphasize the novelty of the present study.
  • The 1st paragraph should be increased by adding information about the prevelance of overweight of Iranian children.
  • How stunting and overweight in children affect their health? Do stunting and overweight be associated with specific chronic diseases at the next stage of children life? The above fact should be described in the Introduction section to reinforce the importance of their study.
  • The 3rd paragraph of the Introduction needs additional analysis concerning the reported previous study in order to emphasize the literature gap that the present study aims to cover.
  • The text in lines 68-74 should be deleted.
  • In line 84, "a face-to-face method" could be changed to "face-to-face interviews".
  • In lines 86-91, the authors should emphasized that these data were self-reported. This fact should also be reported at the end of the Discussion section, as a limitation of the study due to recall bias.
  • The text in section 2.3 should be organized in one paragraph.
  • In section 2.4, the authors should state that the collected data are self-reported.
  • The Tables are not very easily readable.
  • In section 4.3, the authors should note that the self-reported data may lead to recal bias.
  • The Discussion section includes several repetitions that the authors should avoid them.
  • English language editing is highly recommended.
Comments on the Quality of English Language

English language editing is highly recommended.

Author Response

  • Comment 1: [In general, the Introduction section is quite short and needs to be enriched by additional data in order to emphasize the novelty of the present study.]
  • Response 1: Thank you for pointing this out. I added and it is mentioned exactly in [Line 37-42], [Line 46-47], [Line 66-67], [Line 70-74].
  • Comments 2: [The 1st paragraph should be increased by adding information about the prevelance of overweight of Iranian children.
  • Response 2: Thank you for pointing this out. I agree with this comment. Therefore, I added it in [Line 46-47], namely, we mentioned that “In 2019, the prevalence of overweight among Iranian children under the age of five was reported 9% [4].”
  • Comments 3: [How stunting and overweight in children affect their health? Do stunting and overweight be associated with specific chronic diseases at the next stage of children life? The above fact should be described in the Introduction section to reinforce the importance of their study.]
  • Response 3: Agree. I/We have revised this point in [Line 37-42], by mentioning that “Stunting in early childhood has both immediate and long-term consequences, including higher risk of illness, impaired cognitive and physical development, and reduced learning capacity [2]. Over time, it can lead to non-comunicable diseases such as diabetes and hypertension, lower work productivity, and adverse reproductive outcomes [3]. Children who gain excess weight after age two are also more likely to develop obesity later in life, increasing the burden of nutrition-related health issues in adulthood [4].
  • Comments 4: [The 3rd paragraph of the Introduction needs additional analysis concerning the reported previous study in order to emphasize the literature gap that the present study aims to cover.]
  • Response 4:  I emphasized this point in [line 66-67], [Line 70-75], as follows: “These studies are repeated every few years to monitor children's health and inform national policies”, and that “Given the limited evidence on the mediating effects of dietary diversity in the Iranian context, this research provides a novel contribution to the literature. By filling this gap, the study provides a better understanding on the pathway between socioeconomic variables and stunting, aiming at providing deeper insights that can support more targeted and effective nutrition interventions .” This clarifies that we have not only updated previous evidence, but also deepend the analysis, through mediation measurement, which allows for a better understanding of the pathway between SES and stunting.
  • Comments 5: [The text in lines 68-74 should be deleted.]
  • Response 5: Agree, I deleted it. It was from template.
  • Comments 6: [In line 84, "a face-to-face method" could be changed to "face-to-face interviews".]
  • Response 6: I agree I changed it.
  • Comments 7: [In lines 86-91, the authors should emphasized that these data were self-reported. This fact should also be reported at the end of the Discussion section, as a limitation of the study due to recall bias.]
  • Response 7: This was mentioned clearly in the Discussion, as follows: [Line 397-398] which may lead to overestimation due to social desirability bias, as well as recall bias affecting the accuracy of responses [65].
  • Comment 8: [The text in section 2.3 should be organized in one paragraph.]
  • Response 8: Thank you for this important tip, I changed it in [line 127-148]
  • Comment 9: [In section 2.4, the authors should state that the collected data are self-reported.]
  • Response 9: [I added Self-reported in line 154]
  • Comment 10: [The Tables are not very easily readable.]
  • Response 10: [Thank you for the suggestion. We attempted to simplify the tables; however, due to the complexity of the data and the need to maintain clarity, consistency, and coherence across all variables and models, significant restructuring risked compromising the interpretability and connection between the tables. We have, however, made minor adjustments to improve readability where possible.]
  • Comment 11: [In section 4.3, the authors should note that the self-reported data may lead to recal bias.]
  • Response 11: Sure, We added in line 396-397.
  • Comment 12: [The Discussion section includes several repetitions that the authors should avoid them.]
  • Response 12: We fixed them in line 269-276.
  • Comment 13: [English language editing is highly recommended.]
  • Response 13: Thank you for your feedback. We carefully revised and edited the manuscript to improve the English language and overall clarity. We hope the current version meets the journal’s standards.

Reviewer 3 Report

Comments and Suggestions for Authors

General Comment

This manuscript addresses a relevant and important topic concerning the dual burden of malnutrition among children in Iran, using a nationally representative dataset. While the paper is well-structured and supported by sound statistical analysis, several key areas require clarification, improved language, and stronger theoretical framing to meet publication standards.

Specific Comments by Line Number

Title and Abstract

  • Lines 2–3: The title is informative, but consider specifying "2–5-year-old Iranian children" to clarify the population studied.

  • Lines 14–33: The abstract is informative, but the sentence in lines 28–29 "A higher DDS reduced stunting and increased overweight..." is potentially misleading. Later results (lines 237–242) indicate DDS was not a mediator, so this needs revision for consistency.

Introduction

  • Lines 36–65: The introduction is comprehensive but overly descriptive. It would benefit from a clearer research gap. Consider emphasizing the lack of evidence on mediation effects (diet diversity) in Iran to justify your study's originality.

Methods

  • Lines 67–101:

    • Add ethical approval details earlier for transparency.

    • Line 99: “Resulting in a final sample…” — Consider clarifying how missing data were handled (e.g., imputation or case-wise deletion?).

  • Lines 114–136: The description of SES variables is solid, but more justification is needed for the categorization logic, especially regarding grouping of mother occupations into binary (working/nonworking). It oversimplifies potentially important differences.

    • Line 129: Consider citing a reference supporting this dichotomization approach.

  • Lines 137–152:

    • DDS construction is mostly adequate, but cutoff decisions for high vs. low quality diet (Line 151) need a justification or citation. Why only 4 food groups instead of the full 7?

Results

  • Lines 171–202:

    • The univariate results are comprehensive. However, the presentation of odds ratios for many subgroups without visual aids (e.g., forest plots) can be overwhelming. Consider summarizing with graphs or clearer tables in the final version.

  • Line 183: Clarify whether stunting prevalence (5.1%) is nationally weighted or raw percentage.

  • Lines 208–217: The conclusion that “no clear pattern” exists for overweight may be premature. Some trends (e.g., lower overweight in poorer households) deserve more nuanced interpretation.

Discussion

  • Lines 246–254: The section is well-organized into subthemes but lacks critical engagement with contrasting literature, especially regarding the null mediation finding. Why do the authors think DDS did not mediate the SES-malnutrition link, contrary to other studies (e.g., lines 330–332)?

  • Line 249–250: "Number of rooms was not linked to the height of the children" — this finding contradicts earlier univariate results (Lines 183–202). Please clarify.

  • Lines 275–302:

    • You cite studies showing both support and contradiction for occupational status impacts. This is good, but more systematic synthesis would be helpful — e.g., highlighting which contexts (urban vs. rural) or household structures (dual-earner vs. single-earner) these results may apply to.

Limitations

  • Lines 371–391: Good acknowledgment of the main limitations.

    • Line 385: "no index defined to measure wealth" — this contradicts earlier sections where assets, room area, and number of rooms were used as wealth proxies. Please rephrase for clarity.

    • Suggest adding a limitation regarding cross-sectional design precluding causality (not clearly stated).

Conclusion

  • Lines 392–400:

    • The authors could enhance the policy implications. For example, what form might "targeted interventions" take? School feeding? Conditional cash transfers? Parent education programs?

    • The conclusion that no SES factors influence overweight should be nuanced, especially in light of other regional studies showing mixed effects.

Author Response

Title and Abstract

  • Comments 1: [Lines 2–3: The title is informative but consider specifying "2–5-year-old Iranian children" to clarify the population studied.]
  • Response 1: Thank you for pointing this out. The title was revised according to the suggestion.
  • Comment 2: [Lines 14–33: The abstract is informative, but the sentence in lines 28–29 "A higher DDS reduced stunting and increased overweight..." is potentially misleading. Later results (lines 237–242) indicate DDS was not a mediator, so this needs revision for consistency.]
  • Response 2: We have revised the abstract, as follows: “Higher DDS was associated with lower stunting and higher overweight but did not mediate the effects of socioeconomic status.” [line 26-27].

Introduction

  • Comment 3: [Lines 36–65: The introduction is comprehensive but overly descriptive. It would benefit from a clearer research gap. Consider emphasizing the lack of evidence on mediation effects (diet diversity) in Iran to justify your study's originality.]

Response 3: We edited it in line 70-74. We have revised the Introduction to better highlight the paper contribution, as follows: “Given the limited evidence on the mediating effects of dietary diversity in the Iranian context, this research provides a novel contribution to the literature. By filling this gap, the study strengthens the evidence base and provides deeper insights that can support more targeted and effective nutrition interventions.” This clarifies that we have not only updated previous evidence, but also deepend the analysis, through mediation measurement, which allows for a better understanding of the pathway between SES and stunting.

Methods

  • Lines 67–101:
    • Comment 4: Add ethical approval details earlier for transparency.

Response 4: We changed it and put it in line 107-112, as follows: “The database used in the present paper came from a survey that was approved by the Research Council of National Nutrition and Food Technology Research Institute and Ethics Committee of National Nutrition and Food Technology Research Institute (reference code: IR.SBMU.NNFTRI.REC.1396.165). Prior to participating in the survey, written informed consent was obtained from all mothers or tutors of children. The datasets used during the current study are available from MA on reasonable request.”

    • Comment 5: [Line 99: “Resulting in a final sample…” — Consider clarifying how missing data were handled(e.g., imputation or case-wise deletion?).]
    • Response 5: Thank you for your comment. Missing data were excluded. This is clarified in the Methods section to clarify in line 104-105.
  • Comment 6: [Lines 114–136: The description of SES variables is solid, but more justification is needed for the categorization logic, especially regarding grouping of mother occupations into binary (working/nonworking). It oversimplifies potentially important differences.]

Response 6: We have entirely revised this question to clarify and justify the categorization, as follows: “Due to the high proportion of housewives and the small number of mothers across other occupations, maternal employment status was categorized into two groups, namely, working and nonworking mothers [27].  Employees, workers, farmers, and shopkeepers were classified as working mothers, and retirees, students, unemployed individuals, and housewives were considered nonworking mothers. This binary classification aimed to facilitate statistical analysis and ensure adequate sample sizes within each group.”

  • Comment 7: [Line 129: Consider citing a reference supporting this dichotomization approach.]
  • Response 7: IPlease check the answer to the previous comment.
  • Comment 8: [Lines 137–152: DDS construction is mostly adequate, but cutoff decisions for high vs. low quality diet (Line 151) need a justification or citation. Why only 4 food groups instead of the full 7?]
  • Response 8: We explained but add one more sentence too in liie 164-166.

Results

  • Comment 9: [Lines 171–202: The univariate results are comprehensive. However, the presentation of odds ratios for many subgroups without visual aids (e.g., forest plots) can be overwhelming. Consider summarizing with graphs or clearer tables in the final version.]
  • Response 9: Following your idea, forest plots was added for indicating better overview of the unadjusted OR.
  • Comment 10: Line 183: Clarify whether stunting prevalence (5.1%) is nationally weighted or raw percentage.
  • Response 10: Thank you for your comment. The reported stunting prevalence of 5.1% is the nationally weighted percentage, calculated using sampling weights derived from the 2016 national census, adjusted for age, gender, and urban-rural residence to ensure representativeness at the national level. We have now clarified this in the line 194.
  • Comment 11: [Lines 208–217: The conclusion that “no clear pattern” exists for overweight may be premature. Some trends (e.g., lower overweight in poorer households) deserve more nuanced interpretation.]
  • Response 11: Thank you for this valuable observation. We agree that some trends regarding overweight—particularly the lower prevalence in poorer households—warrant a more nuanced interpretation. In the revised manuscript, we have elaborated on these patterns and clarified that while no consistent or statistically significant associations were observed across all SES indicators, certain trends (e.g., inverse relationships with asset ownership or house area) may reflect emerging disparities that should be monitored in future studies. Line 418-420 ….

Discussion

  • Comment 12: Lines 246–254: The section is well-organized into subthemes but lacks critical engagement with contrasting literature, especially regarding the null mediation finding. Why do the authors think DDS did not mediate the SES-malnutrition link, contrary to other studies (e.g., lines 330–332)?
  • Response 12: We appreciate this comment. While previous studies have found diet diversity to mediate the relationship between SES and malnutrition, our null mediation finding may reflect the limited variability in DDS across SES groups in our sample. It’s possible that broader structural factors, such as food access and cultural dietary norms, play a more dominant role in shaping children's diets, weakening the mediating effect of DDS. This underscores the complexity of malnutrition pathways and suggests that further research should explore additional behavioral and environmental mediators.” Line 350-357.
  • Comment 13:[Line 249–250: "Number of rooms was not linked to the height of the children" — this finding contradicts earlier univariate results (Lines 183–202). Please clarify.]
  • Response 13: Thank you for your insightful comment. We acknowledge the apparent discrepancy. In the univariate analysis (Lines 183–202), number of rooms was significantly associated with child height. However, in the multivariate logistic regression, this association was no longer statistically significant after adjusting for other socioeconomic variables. This suggests that the initial relationship observed in the univariate analysis may have been confounded by other factors such as household income or parental education. We have revised the text to clarify this point in the Results and Discussion sections. [Line 260-263]
  • Comment 12: [Lines 275–302: You cite studies showing both support and contradiction for occupational status impacts. This is good, but more systematic synthesis would be helpful — e.g., highlighting which contexts (urban vs. rural) or household structures (dual-earner vs. single-earner) these results may apply to.]
  • Response 14: Thank you for this suggestion. We have now incorporated a review study from Bangladesh showing higher stunting rates in rural areas compared to urban areas, to better contextualize our findings regarding residential differences. [Line 300-303]

Limitations

  • Comment 15: [Lines 371–391: Good acknowledgment of the main limitations. Line 385: "no index defined to measure wealth" — this contradicts earlier sections where assets, room area, and number of rooms were used as wealth proxies. Please rephrase for clarity.]
  • Response 15: We have clarified that assets are used as wealth proxies but that no indication of income was available [Line 397-399].
  • Comment 16: Suggest adding a limitation regarding cross-sectional design precluding causality (not clearly stated).

Response 16: Thank you for your comment. I added it in Line 419-421, as follows: “Given the cross-sectional nature of this study, causality cannot be inferred between socioeconomic factors, dietary diversity, and child nutritional outcomes. To measure causal relationships, longitudinal data should be used, which are not available for Iran”.

Conclusion

  • Lines 392–400:
    • Comment 17: The authors could enhance the policy implications. For example, what form might "targeted interventions" take? School feeding? Conditional cash transfers? Parent education programs?
    • Response 17: We add it in Line 419-421, as follows: “These interventions could include conditional cash transfer programs, housing support, parental education initiatives focused on nutrition and caregiving, and community-based child nutrition services such as daycare or preschool feeding programs.”
    • Comment 18: [The conclusion that no SES factors influence overweight should be nuanced, especially in light of other regional studies showing mixed effects.]
    • Response 18: We add it in line 412-416.

Round 2

Reviewer 2 Report

Comments and Suggestions for Authors

The authors have significantly improved their manuscript.

Reviewer 3 Report

Comments and Suggestions for Authors

Authors have addressed all concerns.